# Magnetic Resonance Imaging Evaluation of Bone Metastases Treated with Radiotherapy in Palliative Intent: A Multicenter Prospective Study on Clinical and Instrumental Evaluation Assessment Concordance (MARTE Study)

**DOI:** 10.3390/diagnostics13142334

**Published:** 2023-07-10

**Authors:** Alfonso Reginelli, Vittorio Patanè, Fabrizio Urraro, Anna Russo, Marco De Chiara, Alfredo Clemente, Umberto Atripaldi, Giovanni Balestrucci, Mauro Buono, Emma D’ippolito, Roberta Grassi, Ida D’onofrio, Stefania Napolitano, Teresa Troiani, Ferdinando De Vita, Fortunato Ciardiello, Valerio Nardone, Salvatore Cappabianca

**Affiliations:** Department of Precision Medicine, University of Campania “L. Vanvitelli”, 80138 Naples, Italy

**Keywords:** magnetic resonance, oncologic, radiotherapy, oncologic diagnostics, bone metastases, lung cancer, breast cancer, prostate cancer, whole-body diffusion weighted imaging, whole body-MRI

## Abstract

Metastasis to bone is a common occurrence among epithelial tumors, with a high incidence rate in the Western world. As a result, bone lesions are a significant burden on the healthcare system, with a high morbidity index. These injuries are often symptomatic and can lead to functional limitations, which in turn cause reduced mobility in patients. Additionally, they can lead to secondary complications such as pathological fractures, spinal cord compression, hypercalcemia, or bone marrow suppression. The treatment of bone metastases requires collaboration between multiple healthcare professionals, including oncologists, orthopedists, neurosurgeons, physiatrists, and radiotherapists. The primary objective of this study is to evaluate the correlation between two methods used to assess local control. Specifically, the study aims to determine if a reduction in the volume of bone lesions corresponds to better symptomatic control in the clinical management of patients, and vice versa. To achieve this objective, the study evaluates morphological criteria by comparing pre- and post-radiotherapy treatment imaging using MRI and RECIST 1.1 criteria. MRI without contrast is the preferred diagnostic imaging method, due to its excellent tolerance by patients, the absence of exposure to ionizing radiation, and the avoidance of paramagnetic contrast media side effects. This imaging modality allows for accurate assessment of bone lesions. One of the secondary objectives of this study is to identify potentially useful parameters that can distinguish patients into two classes: “good” and “poor” responders to treatment, as reported by previous studies in the literature. These parameters can be evaluated from the imaging examinations by analyzing morphological changes and radiomic features on different sequences, such as T1, STIR (short tau inversion recovery), and DWI-MRI (diffusion-weighted).

## 1. Introduction

Bone represents one of the main sites of metastasis among epithelial tumors, with the highest incidence in the Western world. According to information provided by the Italian Association of Medical Oncology (AIOM), approximately 35,000 new cases of bone metastases occur each year in Italy, with the spinal column being the commonly affected site [1]. 

Skeletal bone is estimated to be the first site of metastasis in more than 50% of breast cancers with a systemic disease burden, and about 65–75% of such patients will develop secondary bone localization of disease [2]. 

The types of tumors that are more prone to spread to the bones are breast, lung, prostate, kidney, gastrointestinal tract, and thyroid tumors [3,4]. Georgy et al. [5], Jacobs et al. [6], and Kakhiki et al. [7] have identified varying rates of spinal metastases in cancer patients, ranging from 40% to as high as 70%. Georgy also noted that, annually, around 5% of cancer patients develop spinal metastases. Metastatic bone disease is a frequently encountered and distressing consequence of cancer, often leading to significant morbidity among individuals with advanced-stage cancer, especially those with breast or prostate cancer [8].

Moreover, in metastatic prostate cancer, bone is the main secondary localization of disease in more than 80% of cases [9]. In addition, about one out of three patients with lung cancer or renal cancer are at risk of developing bone metastases [10,11,12]. Normal bone function relies on a delicate balance and collaboration between two types of bone cells—osteoblasts and osteoclasts—that cooperate to ensure continuous remodeling of bone tissue [13].

In bone metastases, the metastatic mass disturbs this balance, leading to a loss of the mechanical properties of bone tissue. Once tumor cells reach the endosteal surface, they stimulate osteoclastogenic activity and create metastases that can have either an osteolytic or osteoaddensifying development. Both developments represent a continuous process that is characterized by regulatory abnormalities of normal bone remodeling processes [9]. Osteolytic metastases occur due to an excess of osteoclastic activity, while osteoaddensifying metastases, such as those seen in prostate carcinoma, arise from abnormalities of osteoblasts and excessive deposition of osteoid tissue [14].

The clinical relevance of a single metastasis is often higher than that of multiple metastases, despite several organs being affected. There are several therapeutic options currently available to manage secondary cancer, such as palliative or curative treatments, depending on the control of the primary neoplasm and the number and location of the metastases. Radiation therapy is often preferred in this context, as it can provide full or partial functional recovery [15]. Pain is the most common symptom of bone metastases, and it can be intense and exacerbated by movements, even minor ones such as coughing, moving limbs, or turning in bed, severely compromising a patient’s quality of life. In some cases, allodynia, or pain perception in response to typically non-painful stimuli, can occur. However, in 25% of cases, metastases can be asymptomatic, and they are often discovered incidentally during diagnostic tests conducted for other reasons [16].

Bone lesions are an important burden for healthcare systems, as they are responsible for a high morbidity index. Injuries are frequently symptomatic and can lead to functional limitations, causing reduction in patients’ mobility and leading to secondary complications, such as pathological fractures, spinal cord compression, hypercalcemia, or bone marrow suppression [17].

Therefore, the treatment of bone metastases involves the collaboration of several professionals, including oncologists, orthopedists, neurosurgeons, physiatrists, and radiotherapists.

Radiotherapic treatment aims to:-control pain, in order to reduce any ongoing antalgic drug therapy and to improve skeletal functions [18,19];-recalcificy osteolytic lesions and prevent the development of “pathological” fractures;-reducing the systemic burden of disease [20];-achieving a “decompressive” effect in spinal cord compression cases (8);-allowing disease control, which is particularly relevant role when bone represents the only site of progression disease in a patient undergoing systemic therapy, as such patients may benefit from locoregional treatment by continuing the current chemotherapy line without having to resort to the next line of treatment [21].

### Study Rationale

Local disease control can be evaluated with both morphological criteria (Recist 1.1) and clinical (symptoms) criteria (with the NRS, WLC-C30, and QLC-BM22 questionnaires) [22,23].

MRI without contrast is a first-choice diagnostic imaging examination because it is well-tolerated by patients and does not expose patients to ionizing radiation or to the side effects of paramagnetic contrast media, thereby allowing an accurate assessment of bone lesions [24].

Furthermore, MRI allows clinicians to overcome the limitations of diagnostic delay, as occurs with CT and bone scintigraphy. This is because the detection of neoplastic cells in bone tissue is not based on the activation of resident bone cells (i.e., osteoblasts and osteoclasts) and the subsequent bone trabceulae histological variations (e.g., sclerosis or lysis). Moreover, unlike PET, MRI is not based on the neoplastic tissue radioactive tracer avidity, which is known to vary according to histology and the stage of disease [25,26]. In addition, MRI provides a complete examination of a patient’s subset, allowing a clinician to distinguish “pathological” bone fractures (i.e., those related to neoplasm bone trabeculae alteration) from “benign” fractures that are attributable to other causes (trauma). Finally, MRI provides a “panoramic” exam that is capable of assessing extra-osseous involvement, especially neurological involvement prior to the manifestation of clinical symptoms [27]. 

One of the secondary aims of this study is to identify potentially useful parameters to distinguish between two classes of patients—“good” or “poor” responders to treatment—as reported by earlier studies in the literature. This distinction could be evaluated from the imaging examinations carried out to analyze morphological changes and radiomic features on different sequences (T1, STIR (short tau inversion recovery), or DWI-RM (diffusion-weighted) (Figure 1 and Figure 2) [24,28,29]. In the future, the analysis of these data could allow further customization of cancer treatment for each individual patient, leading to better-tailored therapy.

## 2. Objective of This Study

The primary objective of this study is to assess the concordance between these two methods of assessing local control, so that a volumetric reduction in bone skeletal lesions corresponds to a better symptomatic control in the clinical management of patients, and vice versa. To accomplish this purpose, the morphological criteria were assessed, comparing morphological imaging (MRI in this case) pre- and post-radiotherapy treatment, according to RECIST 1.1 criteria. Secondary objectives of this study include an evaluation of MRI features—both quantitative features, such as ADC/DWI, and radiomic features. Patient-related clinical parameters (gender, age, and tumor histology) will also be evaluated in accordance with the morphologic and diagnostic imaging features in order to evaluate any response-to-treatment predictive factors. All these parameters will be assessed in the two patient groups—the treatment responders and the non-responders—assessed by instrumental and clinical methods.

## 3. Study Design

A non-profit multicenter prospective observational study will be carried out according to standard clinical practice.

### 3.1. Study Population

From 1 October 2022 to 31 September 2024 (with the time calculated in consideration of the expected accrual for this type of pathology by analyzing the patients treated in the past and the time for company authorization at satellite centers), patients were or will be enrolled consecutively when they met/meet the following criteria:

#### 3.1.1. Inclusion Criteria

Patients with bone metastatic localization;Patients in follow-up and/or undergoing systemic therapy via chemotherapy, target therapy, immunotherapy, hormone therapy, or bisphosphonate therapy, or patients developing any bone metastases requiring radiotherapic treatment;An indication for radiotherapy treatment with palliative intent, using a 3D conformal technique, intensity modulated radiotherapy (IMRT), or volumetric modulated arc therapy (VMAT);The absence of contraindications to MRI imaging; andAn estimated survival prognosis of more than six months.

#### 3.1.2. Exclusion Criteria

Previous radiotherapy in the same bone district;Patient’s refusal to undergo MRI.

A sample of 174 subjects will be necessary to estimate a value of K that is equal to 0.8 (standard deviation 0.3) with a 95% confidence interval, a margin of error of 5%, and a dropout rate of 20%. Thus, approximately 60 patients per center have been or will be recruited.

## 4. Material and Methods

Radiotherapic treatment characteristics (doses, volumes, technique) will be assessed on the basis of a patient’s clinical condition (performance status, assessed according to the Karnofsky Performance Status Scale (KPS) and treatment compliance), the stage of disease and life expectancy, and socio-familial status (family support for treatment and treatment costs). The control of algic symptoms will be assessed through the numeric pain rating scale (NRS), with values ranging from zero to ten, where zero represents the absence of pain and ten represents the maximum expression of pain. Overall survival will be calculated as the time from radiotherapy treatment to death or, in the case of living patients, to the last follow-up date. RECIST criteria 1.1 will be used to evaluate local and distant progression of disease. Local progression will be assessed as evidence of progression of the treated bone lesion, while distant progression will be assessed via the appearance of new bone lesions or progression to other sites (distant progression). Local and distant progression-free survival will be calculated as the time from radiotherapy treatment to local or distant disease progression or, in case of absence of progression, to the last follow-up date. These variables will be collected by descriptive statistics of the enrolled population. A special database has been developed, in which data will be entered on the main characteristics of the patients, the neoplasm, and the treatments being analyzed. The validated questionnaires of the European Organization for Research and Treatment of Cancer (EORTC)), QLQ-C30 (version 3) and QLQ-BM22 will be used in annex 3—the rescheduling of ongoing analgesic therapy. All of the collected data are summarized in Table 1. Therefore, patients will undergo whole-body MRI staging prior to radiotherapy treatment and follow-up 2 and 6 months after the end of radiotherapic treatment. As far as outpatient visits are concerned, as per normal clinical practice, patients will attend at 7 days (these will only be clinical visits, with the administration of questionnaires to assess performance status and pain control), at 2 months after the imaging check, and at 6 months after the new imaging check (or sooner in cases of no pain control).

## 5. Statistical Analysis

The concordance between the treatment response calculated with the pain assessment scales (NRS and EORTC QLQ questionnaire—BM22, EORTC QLC C30, version 3) and the MRI-based imaging assessment will be performed using Cohen’s standardized Kappa coefficient (K). A sample of 174 subjects is required to estimate a K-value of 0.8 (standard deviation 0.3) with a 95% confidence interval, a margin of error of 5%, and a drop-out rate of 20%. Therefore, approximately 60 patients per center will be recruited. Data will be analyzed using SPSS v23.00 software. The secondary objectives of this study are to compare the two patient populations (responders and non-responders) by analyzing the distribution of clinical variables (usually categorical: sex, pathology) and MRI features (usually quantitative variables: DWI parameters, ADC, radiomic features). For quantitative variables, the Student’s t-test will be used, and for categorical variables, the Chi-square test will be used.

## 6. Discussion

Metastatic bone disease is a growing concern among cancer patients, as it severely affects their quality of life [30]. As a result, the accurate staging and evaluation of the disease’s response to treatment have become increasingly important. Unlike X-ray-based imaging techniques such as radiography and CT scans, modern imaging using MRI and PET allows for the detection and evaluation of bone metastases before and after therapy, without the associated drawbacks [31]. One advantage of whole-body MRI (WB-MRI) is that it can identify metastases throughout the body in a single imaging session. MRI provides high sensitivity and specificity for studying the bone marrow, the initial site of landing of bone metastases, and the development of bone metastases [32,33]. Diffusion-weighted sequences, which track water movement within tissues, not only indicate changes in the bone marrow, but also serve as indices of the effectiveness of radiation therapy for bone metastases [34]. MRI can also detect and characterize various types of bone metastases, such as lytic, sclerotic, radio-occult, or mixed [35]. In fact, comparative studies of WB-MRI and PET/CT have demonstrated the former’s superior accuracy in detecting bone metastases in advanced cancer [36,37]. As a result, these methods are expected to see increasing use in clinical practice in the future, with positive impacts on metastatic patients, including better reclassification following therapy and the use of improved and updated therapies based on modern imaging assessments [38,39]. Bone metastases typically indicate a poor prognosis, but recent years have witnessed significant progress in systemic and supportive therapies, which have improved patients’ survival [9]. Most patients with bone metastases require active treatment because of pain, difficulty in walking, pathological fractures, spinal cord compression, hypercalcemia, and neurological deficits that follow [3]. Radiation therapy has been proven to be an effective palliative treatment in metastatic disease, alleviating patients’ pain and enhancing their quality of life [40,41]. MRI has also been found to be a crucial tool in assessing the response to treatment of bone metastases, offering greater accuracy in detecting metastasis than other imaging methods and playing a central role in the analysis of morphological changes induced by therapy [42]. 

Although our study involves a small number of patients, the combined evaluation of signal intensity changes in high b-value DW sequences and mean ADC value has been shown to be a useful tool in detecting skeletal metastases and early therapy-induced effects. However, more large-scale studies are required to obtain more reliable data. If these findings are confirmed, new imaging methods may enable us to predict treatment response soely based solely on MRI-QWI alteration, allowing for early intervention and efficient therapeutic response, which could significantly enhance the quality and life expectancy of patients with metastatic disease, who unfortunately represent a significant proportion of the oncology population.

## 7. Limits

The initial phase of a clinical study is represented by sampling. In order to generalize the results obtained, it is necessary that the sample examined is representative and heterogeneous. This increases the probability of encountering systematic errors (bias) linked to the non-representativeness of the sample produced by the sampling procedure: estimates systematically deviate from the population parameter that one wishes to examine. Another limitation of the study can be represented by the short observation period to which the patients are subjected. This implies that the results of the study are proportional only to the follow-up time of metastatic patients.

## 8. Future Perspectives

Today, bone metastatic disease is a serious concern that impacts the quality of life of an increasing number of oncological patients. The accurate staging and evaluation of disease response to treatments are becoming increasingly important for dealing with this problem. Modern imaging techniques, such as MRI and PET, are particularly useful in detecting and evaluating bone metastases, both before and after therapies. They are also less inconvenient than imaging techniques based on X-rays, such as radiography and CT. 

One of the major advantages of whole-body MRI (WB-MRI) is its ability to detect metastases throughout the body in a single imaging session. WB-MRI provides high anatomical contrast, making it useful for studying the bone marrow where bone metastases first develop. Additionally, MRI can obtain diffusion-weighted sequences that not only reveal changes in the bone marrow but also assess the effectiveness of radiation therapy for bone metastases. It can detect and classify different types of bone metastases, including lytic, sclerotic, radio-occult, and mixed. Comparisons with PET/CT have shown that WB-MRI is superior in detecting bone metastases in advanced cancer, surpassing other techniques such as bone scan (BS) and CT. WB-MRI is expected to become more widely used in clinical practice in the future, resulting in positive effects on metastatic patients. It can help reclassify patients after therapy and guide the use of improved and updated treatments based on evaluations using modern imaging.

Bone metastatic disease is a serious medical condition that can be life-threatening. It is a common progression of different types of cancer, such as prostate, breast, and lung cancer. This condition can cause bone pain, fractures, and growth retardation. Managing bone metastatic disease requires a multidisciplinary approach involving radiation oncology, medical oncology, palliative care, and orthopedic surgery.

Detecting and assessing bone metastases is crucial for managing this condition. Traditional imaging modalities used for this purpose include X-rays, bone scanning, CT scans, and MRI scans. X-rays are useful for evaluating bone density but can only detect bone lesions when they have already destroyed around 30% of the bone’s mineral content.

Bone scanning involves using low doses of radioactive materials to visualize bone lesions. However, it has limitations in terms of sensitivity and specificity. CT scans can also detect bone lesions but are limited in detecting early-stage disease and expose patients to radiation.

MRI scans, on the other hand, do not use radiation and provide more detailed and precise images of affected areas. They are more expensive and time-consuming, and some patients may not be able to tolerate being in the tight, enclosed space of a magnetic tube.

PET scans have also been used for detecting and assessing bone metastases, but they have lower spatial resolution and provide less detailed information about individual lesions.

In conclusion, the detection and assessment of bone metastases in patients with bone metastatic disease are essential for proper management. Imaging modalities such as X-rays, bone scanning, CT scans, MRI scans, and PET scans have their advantages and limitations. The choice of imaging modality depends on factors such as the availability, cost, patient tolerance, and need for detailed information about the lesions.

So, what makes WB-MRI stand out from other imaging modalities? It allows for a highly sensitive, whole-body assessment of bone metastases in a single, relatively short imaging session. Therefore, it avoids the inconvenience of multiple imaging sessions and reduces radiation exposure to patients. This is particularly relevant for patients with advanced cancer or those with multiple metastases in different areas of their bodies. In conclusion, the rapid development of imaging technology has revolutionized the diagnosis and management of bone metastatic disease. MRI and PET have emerged as the most accurate and effective imaging modalities for detecting and evaluating bone metastases. Among them, WB-MRI stands out for its ability to provide a whole-body assessment in a time-efficient and non-inconvenient manner. As technology continues to advance, it is expected that WB-MRI will play an increasingly central role in the detection and management of bone metastatic disease.

## 9. Conclusions

Bone metastases are a frequent complication in advanced cancer cases and can greatly impact the quality of life for patients. These occur when cancer cells spread from their original site and begin to grow in bone tissue. Breast, prostate, lung, and thyroid cancers are some of the most common types that metastasize to bone [29]. When bone metastases occur, they can lead to a range of complications including pain, fatigue, and limitations in movement [5]. Patients with bone metastases are also at a higher risk of experiencing bone fractures and spinal cord compression, which can result in neurological symptoms and even paralysis [24]. Unfortunately, there is currently no cure for bone metastases, so treatment focuses on managing pain and enhancing the patient’s quality of life. Common treatment options include radiation therapy, systemic treatments, and surgery [35]. Radiation therapy is frequently used to relieve pain and improve the functionality of the affected bones. The delivery of radiation therapy can be in a single dose or multiple sessions, depending on the extent and severity of the metastases [6]. Systemic treatments such as chemotherapy, hormone therapy, and targeted therapies are used to shrink or slow down the growth of cancer cells in the bone tissue [38]. Hormone therapy is often employed to treat bone metastases in breast and prostate cancers. By blocking the production or activity of hormones that stimulate cancer cell growth, hormone therapy can effectively manage the condition [20]. 

Chemotherapy, another systemic therapy, is used to destroy cancer cells in the bone tissue. However, it can also cause significant side effects such as nausea, vomiting, and hair loss [19]. 

Supportive care is important for patients with bone metastases in addition to the treatments mentioned. This includes managing pain, providing physical therapy, and offering emotional support [25]. Pain management is crucial as it is a common and debilitating symptom associated with bone metastases [10]. Physical therapy can improve mobility and function, while emotional support can help patients cope with the psychological impact of cancer and its treatment [31]. 

The prognosis for patients with bone metastases is generally poor, with a five-year survival rate of less than 20 percent [41]. However, advancements in systemic and supportive therapies have increased life expectancy for these patients [28]. Early detection and aggressive treatment of bone metastases have been shown to improve outcomes and quality of life [17]. Magnetic resonance imaging (MRI) plays a vital role in assessing the response to treatment of bone metastases.metastases [32]. MRI is a non-invasive imaging technique that produces detailed images of internal organs and structures using magnetic fields and radio waves [18]. 

It is more accurate than other imaging methods in detecting bone metastases and can also analyze morphological changes induced by treatment, such as tumor size reduction or bone sclerosis [22,26]. 

In recent years, there has been growing interest in using MRI to monitor the response to radiation therapy in patients with bone metastases. Radiation therapy is a commonly used palliative treatment for bone metastases that involves the use of high-energy radiation to kill cancer cells and reduce pain [15]. Studies have shown that a reduction in tumor size on MRI after radiation therapy is associated with improved pain relief and quality of life for patients [7]. However, more studies are needed to fully understand the benefits and limitations of using MRI to monitor the response to radiation therapy in patients with bone metastases. One challenge is that changes in bone density and structure can make it difficult to accurately assess treatment response using conventional MRI techniques [9]. Newer techniques, such as diffusion-weighted MRI and dynamic contrast-enhanced MRI, may be more effective in this regard [38]. In addition, the use of MRI to monitor the early effects of radiation therapy on bone tissue may help to identify patients who are at greater risk of developing fractures or other complications. Research has shown that radiation therapy can cause transient or permanent changes in bone structure, depending on the dose and duration of treatment [42]. These changes can lead to a loss of bone density and strength, which increases the risk of fractures [36]. Early detection of these changes may allow for earlier intervention and the implementation of preventative measures, such as bisphosphonate therapy or physical therapy [14]. 

Overall, the use of MRI to monitor the response to radiation therapy in patients with bone metastases is a promising avenue of research that may lead to improved outcomes and quality of life for affected patients. While more studies are needed to fully understand the advantages and limitations of this technique, it is clear that MRI has an important role to play in the management of this challenging and life-limiting condition. With continued research and innovation, we may be able to further extend the lives of patients with bone metastases and provide them with a better quality of life.

## Figures and Tables

**Figure 1 diagnostics-13-02334-f001:**
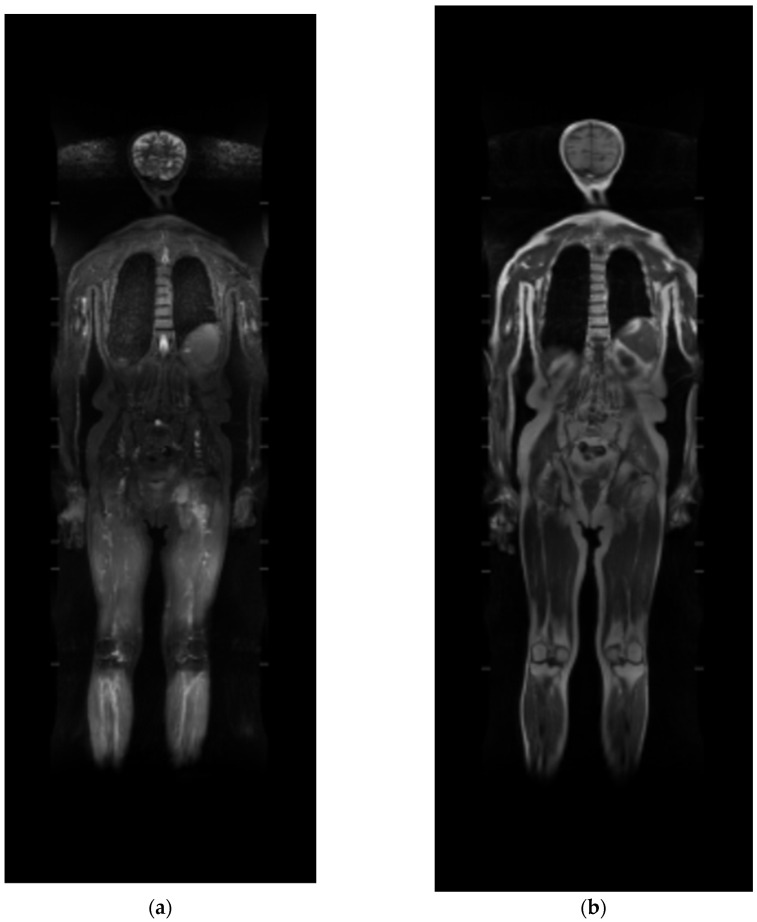
Whole-body MRI: (**a**) panoramic short tau inversion recovery (STIR) sequence and (**b**) T1w sequences obtained from post-processing stitching.

**Figure 2 diagnostics-13-02334-f002:**
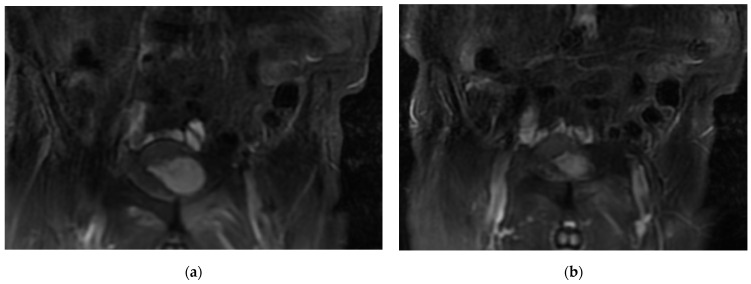
STIR sequence showing right pubic bone metastasis in a patient with bladder cancer (**a**,**b**).

**Table 1 diagnostics-13-02334-t001:** Collected data and reporting system summary. ECOG: Eastern Cooperative Oncology Group; NRS: nutrition risk score; EORTC QLQ: European Organisation for Research and Treatment of Cancer Quality of Life; BMI: body mass index; RECIST: response evaluation criteria in solid tumors; RT: radiotherapy; IMRT: intensity modulated radiotherapy; VMAT: volumetric modulated arc therapy.

Scope	Data
Biographical data	Sex, age, date of recrutiment
Medical histoy	Comorbidity, current antalgic drug therapy
General evaluation	ECOG performance status, pain status accordind to NRS and EORTC QLQ questionnaire—BMI
Disease characteristics	Histology, date of diagnosis, secondary bone disease, date of onset of symptoms
Treatment	Possible chemotherapy/target therapy/concurrent or sequential immunotherapy/hormone therapy
Response to therapy	Instrumental: according to RECIST criteria 1.1 Clinical: according to specific questionnaires
Radiotheapric treatment	Conventional RT/IMRT/VMAT radiothetrapy (technique, total dose, dose per fraction, volumes, start/end date)
Other clinical outcomes	Patient status (alive, deceased), relapse status (local or distant progression y/n), local and distant disease progression free survival.

## Data Availability

The data presented in this study are available on request from the corresponding author. The data are not publicly available due to privacy and ethical restrictions.

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
