# Peer review of "Magnetic Resonance Imaging Evaluation of Bone Metastases Treated with Radiotherapy in Palliative Intent: A Multicenter Prospective Study on Clinical and Instrumental Evaluation Assessment Concordance (MARTE Study)"

_diagnostics, 2023, doi:10.3390/diagnostics13142334_

Round 1
Reviewer 1 Report
Bone metastasis is a very common secondary malignant bone tumor in clinical practice, especially in recent years, with the continuous improvement of relevant clinical treatment measures, the survival time of malignant tumor patients is gradually extended, leading to an increasing number of metastatic tumor patients. How to accurately evaluate the treatment efficacy of patients with disease is a concern for clinical doctors. This design aims to evaluate clinical efficacy through changes in MR imaging before and after radiotherapy, which is a very good research approach and method. Through the author's research work, it is possible to achieve the research goals set by the author.
Author Response
Dear reviewer,
first of all i want to thank you for your effort.
I'm really proud you appreciate our protocol.
Our hope is, once this protocol starts, is to improve always more and more oncological patients' quality of life and to ease clinical managment of them.
thank you sincerely,
vittorio
Reviewer 2 Report
Thank you for inviting me to review the manuscript entitled „ Magnetic resonance imaging evaluation of bone metastases treted with radiotherapy in palliative intent: a multicentre prospective study on clinical and instrumental evaluation assess-4 ment concordance (MARTE study)” submitted for potential publication in the journal of Diagnostics. The work is interesting and in the scope of the journal.
The objectives of the study are presented inconsistently. The primary aim is described in line 108. The secondary objectives are presented in the introduction in line 130. Please move them to the objectives section. The introduction should include only a presentation of the research question and justification for undertaking the study.
The protocol should have the structure recommended by the SPIRIT checklist. There is no need to provide discussion, limitations, or conclusions as this is not a study. Also, some elements that are mandatory, have not been included in this protocol – please kindly consult the checklist.
Was the study registered? If not yet, please provide the name of the intended trial database.
There are plenty of typos, for example, Line 6 - a double comma at the end of the line; in Table 1 – History and Radiotheapric, in the title – treted, line 162 – radiothrerapy, line 167 – Radiotherpay,
Please kindly explain the abbreviations under Table 1.
Please kindly check the capitalization.
American English is mixed with British English, please harmonize.
Author Response
Thank you for inviting me to review the manuscript entitled „ Magnetic resonance imaging evaluation of bone metastases treted with radiotherapy in palliative intent: a multicentre prospective study on clinical and instrumental evaluation assess-4 ment concordance (MARTE study)” submitted for potential publication in the journal of Diagnostics. The work is interesting and in the scope of the journal.
The objectives of the study are presented inconsistently. The primary aim is described in line 108. The secondary objectives are presented in the introduction in line 130. Please move them to the objectives section. The introduction should include only a presentation of the research question and justification for undertaking the study.
This is due to make more fluent our paper, anyway your correction has been accomplished.
The protocol should have the structure recommended by the SPIRIT checklist. There is no need to provide discussion, limitations, or conclusions as this is not a study. Also, some elements that are mandatory, have not been included in this protocol – please kindly consult the checklist.
Dear reviewer, we initiallu structured this protocol paper following instruction of SPIRIT checklist, but after the first formal editorial review, Dr. Moon Zheng via email in May 2023 suggested us to reach at least 4000 words and then to structure the paper as the usual paper formatting. This is why our paper is not structured following the SPIRIT checklist. In order to find a common point of view we tried to improve our protocol with some missing informations and then we’d like to insert the SPIRIT checklist as additional appendix to the original manuscript.
Was the study registered? If not yet, please provide the name of the intended trial database.
The study has been registered and this missing information has been added to the manuscript.
There are plenty of typos, for example, Line 6 - a double comma at the end of the line; in Table 1 – History and Radiotheapric, in the title – treted, line 162 – radiothrerapy, line 167 – Radiotherpay,
Typos has been corrected.
Please kindly explain the abbreviations under Table 1.
abbreviations have been explained.
Please kindly check the capitalization.
Capitalization has been checked
American English is mixed with British English, please harmonize.
Although American and Biritish are basically the same language, I understand there were some passage with some imperfections that we tried to harmonize.

Round 2
Reviewer 2 Report
Thank you for clarifying the editorial office's approach to the paper's structure and length. I appreciate that you added the checklist and tried to follow the recommendations included. I don't have any additional comments.
Minor language corrections can be done editorial office.